# Risk Implications for the Role of Budgets in Implementing Post-Acquisition Systems Integration Strategies

**Nazila Razi** [1,*]**, Elizabeth More** [2] **and Gensheng Shen** [3]

1   Faculty of Postgraduate Accounting, King's Own Institute, Sydney 2000, Australia
2   Chief Academic Office, Study Group, Sydney 2010, Australia; EMore@studygroup.com
3   King's Own Institute, Sydney 2000, Australia; shen@koi.edu.au
*   Correspondence: nazila.razi@koi.edu.au

**Abstract:** This paper studies the role of budgets in implementing the systems integration strategies in an Australian post-acquisition case of two organisations and reducing its associated often-regarded high risks. It attempts a fresh narrative approach to examine the evolution of accounting and its effects on the challenges of post-acquisition integration processes by using the performative approach such as the sociotechnical networks of Actor Network Theory in a broader analytical framework as a possible solution to reducing the risks inherent in systems integration. The methodology of the case study is based on Callon's model of Four-Moment translation where integration strategy and budgets are regarded as social practice and defined relationally as bundles of activities and take form in and through practice and interaction between diverse actors and actants. A qualitative approach is adopted in the examination of the systems integration networks in an Australian post-acquisition case. Data was collected and analysed using semi-structured interviews. It was found, through the examination of the routine practices of systems integration strategy making and how people enact and draw on a certain financial report on a daily basis to perform systems integration network strategies, that material forms of accounting act as a powerful structuring and inscription tool in integration activities, thus shaping integration strategic options and post-acquisition economic conditions of the organisation. The result shows how the risk could be reduced in the post-acquisition system integration. The research contributes to the risk, change, and accounting literatures by providing insights into the mundane and ordinary practices of different aspects of integration strategy making, and the way employees enact and draw on accounting numbers on a day-to-day basis to perform systems integration network strategies. This case study facilities this research to be further developed and broadened in terms of other cases, industries, and countries.

**Keywords:** systems integration networks; budgets; actor network theory; human and non-human actors; post-acquisition risks

## 1. Introduction

Acquisitions are seen as an important growth strategy and are commonly used by corporate executives and practitioners for business expansion purposes. Continuing research on strategic integration in the post-acquisition process is due to the rich literature reporting the failures of most acquisitions and mergers world-wide, including in Australia. For example, Yaw (2016) reported that "M & A failure rate is very high; averaging about 50%, regardless of the initial high hopes" and also noted (p. 1.) "the integration stage as one of the critical stages within the whole M & A process which can contribute immensely to M & A failure".

The year 2015 witnessed the highest ever global merger and acquisitions (M&A) activity (Reuters 2016) worth US $4.5 trillion. In 2018, Australia experienced 1900 M&As, with a value of approximately $185 billion with many of them failing because risks were not fully understood (YouLegal 2019). For instance, Wesfarmers acquired Coles in 2007 for

$21b. But, recently Wesfarmers demerged from the Coles Group and both are back where they started. So, it is important to study this in the Australian context. Despite high volumes of this activity, it is clearly high risk, with more than 70% of the deals failing to create value (King et al. 2004; Marks and Mirvis 2011; Christensen 2016).

A variety of factors have been identified for this high failure rate; however, successful systems integration strategies are the most significant contributors to the realisation of acquisition value. Some studies show up to 60% of deals that anticipated benefits were directly dependent on this process (Curtis and Chanmugam 2005; Lin et al. 2010; Tanriverdi and Uysal 2011; Bajraktari 2016). However, the post-acquisition system integration is still regarded as one of the most difficult challenges and continues as a common cause of acquisition failure (Posnick and Schoenborn 2007; Henningsson et al. 2018; Rouzies et al. 2019). "Differences in systems and processes can make the business combination difficult and often painful right after the merger" (Investopedia 2021), and hence, the factors that assist with effective systems integration strategy are of paramount importance. Given most studies and reports predict a rise in global merger and acquisition activity in the coming years (Baker McKenzie Global Transactions Forecast 2018), there is a greater need for improved systems integration strategies and expanding knowledge of factors with the potential to facilitate it and reduce risk of failure. Could accounting be amongst those factors assisting an effective systems integration strategy and overall acquisition performance?

The literature reveals sizable studies on the link between accounting and strategy in general, but there is a scarcity of empirical research that examines specifically the role of accounting in integration strategies and processes (Hitt et al. 1990; Yazdifar et al. 2008; Capece et al. 2017; Razi and Garrick 2019). What if accounting plays a vital role in the systems integration, assisting the management of Systems Integration Networks (SIN) complexities? If so, accounting could be instrumental (perhaps substantially) in increasing the benefits of an acquisition and reducing its risks. This could have practical implications for managers engaged in acquisition activity and, consequently, worthwhile investigation.

This case study research examines the role of budgets in post-acquisition systems integration[1] strategies. The study provides an outline of how the researcher participated in observing an Australian company's systems integration activities, following the acquisition of another smaller company that was integrated with the acquiring company's operational system (Pronto), and how its employees had to adopt this new technology that they had not used previously.

According to Latour (1986, 2005) and Callon (1986b), the classic, practice-based approach is where strategy and budgets are regarded as social practice, defined as relationally understood as the bundles of activities that comprise interaction between diverse actors and actants. Drawing on the principle of 'action in making' actors, both, human and non-human (i.e., budgets) factors are followed and documented because they do act in their everyday business of strategising SIN relations.

The study adopts an Actor Network Theory (ANT) as an analytical framework, because it is particularly well adapted to the study of the role played by science and technology in structuring power relationships. Furthermore, 'Actor Network Theory' focuses on connecting intention with objects of the acquiring company and this is very useful when considering how actors influence the successful implementation of technology change' (Bruun and Hukkinen 2003, p. 104). In line with the chosen analytical framework, the notion of translation is well suited, since, according to Callon (1986b), it reveals all the mechanisms and strategies through which an actor identifies other actors or elements and connects them with one another (p. 193).

The study illustrates the performative powers of budgets, which make SIN relations strategy visible and actionable. They structure and control the systems integration strategies of the firm, help with risk management, and play an active role in integration strategy formulation, constrain its implementation, and configure the identity of strategic actors, leading to the integration strategy confronting strategic change and risk issues.

This study makes contributions to the research in the fields of risk, change, and accounting, as it provides further knowledge on the important role that budgets play in forming integration strategies and how they are implemented. It also contributes to the strategic management literature by empirically demonstrating how integration strategy is defined relationally, and by proposing changes to the theoretical base of management accounting by empirically illustrating the agency of budgets as a non-human actant in strategising SIN relations.

The implication for practice is that it provides an opportunity for the accounting profession to reflect on the integration skills required by accountants and the main factors on which they should focus to achieve timely and effective integration. Business executives engaged in acquisition activities are directed to consider budgets as an important part of performative action in the formation and implementation of SIN relation strategies, in order to have a less risky and more successful acquisition.

The paper is divided into five further sections. First, a review of the key literature occurs, addressing evaluation of ostensive versus performative studies in the accounting strategy research field, and risk aspects. Second, a review of the link between Actor Network Theory and SIN is discussed. Third, we provide the context of the case study. Fourth, a discussion of empirical observations precedes the conclusion in the fifth, Section outlining the implications of this research.

## 2. Literature Review

### 2.1. Studies on Management Accounting and Acquisition

A literature review demonstrates limited research examining the link between management accounting and post-acquisition integration strategies, with only a handful of examples (Moilanen 2016; Kyriazopoulos and Logotheti 2019). Looking historically, according to Hopwood (1973), management accounting systems in the organisations are the source of formal information, and Mumford and Pettigrew (1975) suggested that possessing such information provides a source of power. Therefore, management accounting systems are the means for distributing power and forming an integral part of the organisation's structure and processes (Jones 1985, p. 89). Their importance stems from the "ability to facilitate organisational integration, to motivate, to assist decision-making, and to provide measurements of performance through enabling characteristics such as the delegation of authority, communication of objectives, participation and informational feedback" (Jones 1985, p. 179). Studies such as those of Roberts (1990), Jones (1985, 1986) and Granlund (2003), further emphasised the importance of management accounting systems after acquisition, since they have the capability to deal with risk and facilitate control and integration in the post-acquisition period.

### 2.2. The Ostensive View of Accounting and Strategy

The number of studies that examine the link between accounting and strategy, in general, are large in size (Boedker 2010; Aaltola 2019; Rashid et al. 2021). Much research in earlier days used contingency theory and portrayed a fairly static view of the relationship between strategy, structure and management controls (Jones 1986; Roberts 1990; Chenhall 2003). Overall, in these studies, the strategy is identified in three main perspectives (Tryggestad 2005). Firstly, strategy as rational plans (Porter 1980), when business strategy is seen as residing in a set of rational plans for the long term, with strategy itself regarded as a formal, deliberate, reasonably rational process that is the domain of a small group of elite actors (see Anthony and Dearden 1980; Porter 1980). A second perspective contrasts with the first, with strategy seen as emergent plans (the classic work of Mintzberg 1988). In this perspective, it is argued that strategy often emerges through processes of negotiation and interaction among organisational actors. And third, is the view of strategising as practice (Whittington 2006), where strategy is considered to be a social process that is enacted through a variety of organisational routines and practices.

In most of these studies, there is a divide between social and economic elements when investigating integration processes, and social considerations seem to be almost excluded. This assigns a secondary function to accounting (Boedker 2010), where it is regarded as an output of strategy, and narrowly focuses on certain factors. This limits the ability of such studies to provide a thorough analysis of post-acquisition integration, since social and economic elements combine in an integration process. Furthermore, the nature of change in integration strategies challenges the ability of these perspectives to explain the agency through which they are formed and persist (Skærbæk and Tryggestard 2010); is limited to the study of changes in management controls that take place after the acquisition, and fail to explore the possible role controls might play in the processes of the creation of integration relations.

### 2.3. The Performative View of Accounting and Strategy

The literature reveals that some recent research calls for a more 'practice-based perspective' on the accounting-strategy relationship (Chua 2007; Guthrie and Parker 2017; Jansen 2018), leading to an increase in our understanding about the constitutive powers of accounting. In these studies, strategy is considered to be a social process (Skærbæk and Tryggestard 2010) that is enacted through a variety of organisational routines and practices (Whittington 2006), necessitating a special consideration of social elements (Roberts 1990). In this view, accounting and financial reports are seen as a practical activity and strategy as a verb (i.e., strategise). Accounting not only controls strategy but also has the potential to create unexpected outcomes (Davila 2005; Razi and Garrick 2019), resulting in new strategy proposals (e.g., SIN). Strategy via accounting is recognised as a social practice (Razi and More 2017), concerned with the everyday actions of strategy makers in an organisation.

This view provides new and interesting ways to comprehend accounting, because accounting technologies, such as budgets, are now considered as central to the sociological implications of accounting and accountants' activities (Justesen and Mouritsen 2011). In contrast, researchers from the 'ostensive' stream assume that strategy and accounting exist as reasonably coherent objects, already knowing the essence of 'accounting' and 'strategy' beforehand. What 'strategy' is, what its properties are, and how it moves and operates in practice is generally understood. Researchers study 'accounting' and 'strategy' as 'objects' by assuming they have pre-determined and fixed forms. According to the performative approach as Latour (1986) suggests:

> " . . . The nature of society is negotiable, a practical and revisable matter (performative), and not something that can be determined once and for all by the sociologist who attempts to stand outside it (ostensive)" (p. 264).

And, as noted by Czarniawska (2001, p. 256):

> " . . . A performative perspective means that the understanding of things (including words) depends upon their use".

So, it is proposed that relationships need to be studied in order to study objects. To do this, it is necessary to, first, investigate how the 'objects' move and are performed in practice, and then, how they take on form in and through relationships and actor networks. Researchers should study 'accounting' and 'strategy' as a fluid relationship of interrelated actors, human and non-human, not as 'black boxed' objects. The performative theory elaborates on the tenets underpinning this approach. As Latour (1986) states:

> "Social phenomena emerge in and through networks of practice, and their identity resides in neither an individual nor a technology but in a chain of relations between actors" (p. 273).

In the studies that take the performative approach (such as Skærbæk and Tryggestard 2010), power is seen as an outcome of situated social processes and, therefore, the researcher should carefully examine the processes that give rise to power as an effect (Murdoch and Marsden 1995). These findings are important in providing new insights as they

assign a central role to accounting and demonstrate that management controls play a performative role (Skærbæk and Tryggestard 2010; Dumay and Rooney 2016) in relation to strategizing in general. Moreover, they suggest that material forms of accounting (such as budgets) not only are the means of representation of an organisation, but also effect organisational changes (e.g., post-acquisition integration), by producing new sets of information as engines to create new actions and resulting practices (MacKenzie 2006; Sorescu et al. 2007; Boedker 2010).

According to the study researcher's knowledge, little investigation occurs to examine whether this is the case in the post-acquisition integration circumstances. Integration research, from a management accounting point of view, is still inadequate due to the existing knowledge being disjointed on different but related aspects of the post-acquisition change integration process. There is little knowledge about the true impact of management controls on post-acquisition integration and acquisition performance. Yazdifar et al. (2008) emphasized the importance of management accounting researchers to further their knowledge by examining parent-subsidiary dynamics, their impact on integration, and how controls are used in these dynamics. Integration processes might be advanced or hindered by controls (Sorescu et al. 2007), so the overall acquisition success will be impacted. Despite the significant contribution of prior studies, the outstanding gap is that little knowledge has been gained on accounting processes in action (Burchell et al. 1980; Granlund 2003; Skærbæk and Tryggestard 2010; Dumay and Rooney 2016) in the creation of integration relations. It seems that existing research inadequately investigates the evolution of management controls and effects on integration processes, thus limiting findings useful to reducing risk in such change processes. Taking a performative perspective, the current case study attempts to fill this gap by examining the "performative role of budgets in the systems integration strategies". It applies Actor Network Theory and the notion of performative theory, proposed by Latour (1987) and Callon (1986b), as a conceptual framework.

*2.4. Risk Management View of Management Control System*

Mergers and acquisitions are risky in business operations due to many obvious and significant differences between the acquiring company and the acquired company, often including culture, processes and systems. The process of applying various approaches of management control systems in the post-acquisition integration itself represents easily many real risks when most cases of the management control systems in post-acquisition integrations often fail due to either the over-emphasis of the Ostensive Approach to using non-human instruments of management control systems in post-acquisition integration, or the one-sided emphasis of the Social Approach (the Performative View) to using human-based instruments of management control systems in post-acquisition integration. Risk is a form of uncertainty in the post-acquisition integration process where some relevant and essential factors are ignored in the actual implementation of management control systems in post-acquisition integration.

The risks of the Ostensive Approach in causing the failure of post-acquisition integration are multi-dimensional but primarily because in this approach, most social considerations seem to be almost excluded (Boedker 2010), plus the ability of this approach is limited in the development of post-acquisition integration. It is perceived as the integration design risk for management control systems and the approach is likely to fail to explain the agency involved in the integration nature of social and economic factors in operation. This is perceived as the operational risk for management control systems (Skærbæk and Tryggestard 2010). Such an approach prevents the study of the changes in management controls that take place after the acquisition and the exploration of the possible role of management control systems in the processes of the creation of integration relations. This is perceived as the management control risk.

The risk of the Social Approach (the Performative View) in causing the failure of post-acquisition integration is clear in lacking the recorded observable processes of the fluid human relationships in management control systems. This is perceived as the experimental

risk, as Skærbæk and Tryggestard (2010, p. 41) pointed out, claiming "power is seen as an outcome of situated social processes" that give rise to power as an effect of management change and control (Murdoch and Marsden 1995). This approach, however, directs us to think of the performative role of the material forms of accounting (such as budgets) in management control systems in post-acquisition integration (MacKenzie 2006; Sorescu et al. 2007; Boedker 2010).

Inspired by the concepts in the paper "The role of accounting devices in performing corporate strategy", (Skærbæk and Tryggestard 2010), the work of Latour (1987) and Callon (1986b) and its underlying Actor Network Theory, this paper develops an holistic approach that integrates all factors, both human and non-human, economic and social, human and technical, thus minimising various risks in the practice of management control systems in post-acquisition integration. The implication for the role of the Actor Network Theory approach to the effective management of risks in the process of management control systems in post-acquisition integration is important and worthy of further research.

## 3. Conceptual Framework

The resolution of systems integration involves a number of networks of heterogeneous actors in more or less stable associations (Law 1992), formed over time through a process of translation (Callon 1986b). Actors considered within the network may be human or non-human and act, or make a difference, within networks (Lowe 2001) that change in an ongoing process of making and remaking (Callon 1986b). Callon and Latour (1981) explain:

> "By translation we understand all the negotiations, intrigues, calculations, acts of persuasion and violence, thanks to which an actor or force takes, or causes to be conferred on itself, authority to speak or act on behalf of another actor or force..." (p. 279).

Therefore, the focus should be placed on the actors with the power to introduce and mobilise other actors and resources (Callon 1986b). In terms of Latour's Actor Network Theory, these powerful actors create an obligatory passage point for the translation processes that follow (Callon 1986b). With the aims of this paper from the perspective of Actor Network Theory, the object of the research is the SIN with reference to the exploration of network representation, actor relations, and the obligatory passage point used to shape those relations. Thus, SIN is a consequence of the observed struggles of actors to translate the framing represented by the new system to address the issue encapsulated within an obligatory passage point formed by a powerful actor. Translation refers to the process by which an actor joins a network, where the powerful actor displaces the interests of others and speaks on their behalf, promoting their own interests as those of the collective (Callon 1986b). This process leads to the construction and stabilisation of the SIN of heterogeneous elements (Callon 1986b; Latour 1999). Callon (1986b) describes the moments of the translation conceptual framework as:

> "The development of an actor-network through four steps of problematization, interessement, enrolment and mobilization, as steps, or moments, constituting the different phases of a general process, where the identity of actors, the possibility of interaction and a process of negotiation is used to enrol the actors into an actor-network" (p. 203).

The different phases are as follow: (1) In the problematization phase as the First Moment, a focal actor strives to become indispensable to other actors by defining the problem and solution if the actors negotiated the obligatory passage point. (2) Interessement as the Second Moment, relates to the activities where the focal actor aims to impose and stabilize the other actors. (3) Enrolment as the Third Moment, is the creation of a sociotechnical network of alliances, to build up agreement among differing actors where they accept their refined roles, given to them to ensure the problem is addressed. According to Latour (1986) this involves: *"Enrolling both social and material actors and actant in new networks"* (p. 275).

Once agreement between actors is reached, commitments should be recorded into the shared memory of the social system (i.e., stabilized) through inscription (Callon 1986a):

> "Entities that comprise networks are often converted into inscriptions or devices such as documents, reports, academic papers, models, books, and computer programs" (Callon 1986a, p. 68).

At this stage, weakness and strength of the bonds between elements are strong determinants of the power and survival of any network (Callon 1986b, p. 4) Mobilisation as the Fourth Moment, relates to the methods used by the focal actor in continuing to speak on behalf of the network. These four moments will be described in the subsequent sections.

## 4. Research Site and Methodology

This study examines an Australian company. The study of Australian cases of acquisitions and mergers is motivated by a number of factors. A crucial one is from a methodological perspective. Choosing the Australian context meant the researcher being able to do the face-to-face interviews and other in person research that was deemed necessary for the study envisaged. Additional factors included the nature of Australia's small open economy with a long history of both attracting foreign investment and investing overseas (Australian Productivity Commission 2020), allowing study contributions in advancing Australia's economic development. Moreover, the Australian focus was designed to facilitate future comparative studies using the interesting work of Karagiannidis (2010) on the synchronisation of fluctuation in trends of acquisitions and mergers between Australian, USA, and UK.

### 4.1. The Case Study Company

Australian Stock Exchange (ASX) listed companies that were engaged in M&A activities were closely examined, and potential candidates for the purpose of this study were identified. Among those, a packaging company (PROC[2]) as a manufacturer and distributor of packaging products was chosen for this study. This particular sample was chosen because this company made an interesting case study for the purpose of this research and represented a significant part of Australian industry that has grown even more in recent years. This Australian company was constantly engaged in acquisition activity (making 17 acquisitions over a period of 8 years) and heavily used budgets as the main means of control in the organisation. It acquired the targeted company in this period and integrated the organisation into its systems. The process took 18 months (a year longer than initially planned), many challenges were faced throughout and, at the end, the expected financial gains were not fully realized, demonstrating the risks involved in such acquisition and change processes. This was largely because, during due diligence, the requirements and practical realities of the crucial systems integration had not received adequate consideration, and problems arose that ought to have been foreseen but were not as some key information was overlooked.

The methodology in this research is a case study. Inscriptions[3] (Robson 1992) are utilized for data analysis of the case. The inscriptions and its associated data in this research consist of field notes, interview recordings and transcripts including primary documents such as annual reports, regional business plans, minutes of executive meetings, office memos, press releases, departmental meeting minutes, observations, the integration cost budget, the acquired company sales report, and notes of corridor discussions with staff and the Managing Director as observed.

The main body of the data was collected through semi-structured interviews (Appendix A provides the Interview Guide) agreed by the organisations involved lasting approximately for one hour each. Crucially, all individuals who were directly involved with the acquisition and integration activity (20 in total) were interviewed without exception. This enabled the researcher to gain richer data than perhaps other methods might have afforded. Interviewee profiles include: Chief Executive Officer, Managing Director, Chief Financial Officer/Company Secretary, National Human Resources Manager, National Compliance and Logistic Manager,

General Manager Procurement, State Manager, Divisional Manager (Ex Sales Manager in the acquired company), National Marketing Manager, Internal Sales Manager, Financial Controller, Warehouse and Logistics Manager for NSW, Marketing Executive, Sales and Supply Chain Manager, New Product Development Manager, Warehouse Staff members, Sales Department Staff, and SME Internal Sales Supervisor.

The strength of semi-structured interviews in case studies is that they are targeted directly on the topic and provides insight into a phenomenon by implying causal inferences (Yin 2003). The interviews were conducted with employees of the acquired and acquirer companies involved (prior, during, and post) in the acquisition activities, with descriptions sought as to how they were 'acting out' everyday life and making integration strategies. In exploring SIN from an ANT point of view, the issues of generalised symmetry, generalised agnosticism, and free association, were identified by Callon (1986b, pp. 198–201) and should be acknowledged. Generalised symmetry highlights a theoretical problem, when sociologists argue over a particular integration strategy, due to that fact that they rarely agree among themselves (Callon 1986b, p. 198). According to Callon (1986b):

> "The researcher needs to convey a narrative that convinces the reader the story line chosen adequately represents the 'technical and the social aspects of the problem studied" (p. 200).

Hence the narrative style of this paper. The principle of 'agnosticism' is where interests of actors are multiple and unspecified (Law 1992) and, hence, researchers should be talking of multiple interests rather than a single interest (Law 1992).

In this study the researcher entered the field sometime after the initial systems integration had taken place. The researcher had neither input nor involvement with the integration process, so was an impartial observer at all times. Every actor's point of view was heard, and none was privileged over the others. The statements made by the actors indicate that the systems integration network was fragile and risky at times, since different actors demonstrated various epistemological beliefs and shared different interests. The researcher witnessed how various actors developed their own understanding and questioned the integration efforts, and, in the meantime, adopted different inscriptions based on how they interpreted the integration. At every stage of the struggles in the creation of SIN, importantly different various elements were allowed to freely associate without any definite boundary between them. Despite the differences, the network finally accomplished its objectives as the Managing Director demonstrated capability in aligning the actors' interests with hers. As a result, association between actors was strengthened and the SIN network stabilised.

The principle of 'free association' is what Callon (1986b) refers to as 'a methodological problem with scientific research' (p. 197). The researcher needs to consider the actors' identity and the significance of their position in the systems integration network. Callon (1986b) asserts that:

> "Science and technology are dramatic 'stories' in which the identity of the actors is one of the issues at hand. The observer who disregards these uncertainties risks writing a slanted story which ignores the fact that the identities of actors are problematic" (p. 197).

To overcome these problems, Callon (1986b) strongly suggests that researchers refrain from making a distinction between natural and social events as the boundary is analytical.

### 4.2. Integration Efforts

A few months after the acquisition, the actual physical integration took place, where four disparate New South Wales state operations were relocated, consolidated and integrated into a single unit of operation. However, the team decided to run their systems standalone from that location for a while, so, the initial budgets and costs were prepared as if they were a standalone business. These budgets and costings were used to determine the approach to integration and, consequently, they tried to keep intact all the systems, process,

procedures, and staff because they had the knowledge of how to operate the business. The intention was not to rush in and try to make a lot of changes, but rather to get to know the people and their methods of operation, and then work out integration strategies. This was because they believed the information provided during due diligence was not sufficient to plan for the integration straight away.

### 4.3. Initial Stages of the Systems Integration Network

The first moment of a translation process is problematization. During this moment, the focal actor initiates the process by identifying other actors with their interests consistent with their own, and then frames the problem in its own terms, and highlights how the problem affects the other actors. At this stage, every actor becomes equipped with an individual identity that is made up of interactions and negotiations with other actors (Callon 1986b). The focal actor then outlines broad strategies to address the problem at hand and establishes an obligatory passage point (OPP) so as to render itself 'indispensable' (Callon 1986b, p. 204). The focal actor's intent is to " . . . *oblige an entity to consent to detour"* (Callon 1986b, p. 26) such that resolution can only be reached through this obligatory passage point (Dumay and Rooney 2016).

The following sections provide a description of the case study with comments. Once the acquiring company's top management concluded that systems integration was vital to achieving the benefits of the acquisition, they pursued the rapid problematization of the integration concept with the actors. Migrating all the business under one single IT system (Pronto) was at the top of the list of the tasks. The National Marketing Manager described Pronto as:

> " . . . Is an ERP system; we do a lot of invoicing from there, all our budgeting and reporting, all our customer relationship management, sales. It is like a bit of everything. It's a one system you use it for everything".

According to the Managing Director:

> 'Pronto is a very strong control platform . . . we needed to integrate them into our system purely for us to have control over the operation of what they [acquired company] were doing . . . All [acquired companies] have to move to Pronto, in fact, the biggest part of the integration is that IT moves them into Pronto . . . ".

So, participation of the IT team and Head was crucial. The CEO described this:

> "IT has a lot of challenges in this [systems integration], everything has to be done now, now and then, there are lean times required. They have a lot of challenges in trying to get it happening really quickly, and bypassing certain lead times. You just have to work to what is needed, sometimes it can be done and sometimes it cannot".

At that stage, the Managing Director took the role of the speaker as the focal actor and communicated the issue to the actors and became responsible for making the acquiring company more efficient and effective through the integration initiative. As the initial step, the Managing Director arranged a first strategy meeting and invited the management team, including the IT department head. As a result, the problematization of systems integration network was expressed by the Managing Director as follows:

> " . . . Their [acquired company] system called Exonet which is a version of MYOB, a terrible system! from inventory controls and management . . . nowhere close to what we have in Pronto" and "All [acquired companies] have to move to Pronto and use it in daily basis . . . ".

This is the obligatory passage point (OPP) for all actors in the network and a challenge to which they must find the solution (Callon 1986b, p. 206).

As the problematizing actor, the Managing Director needed to encourage other entities in the systems integration network to recognise the link between the need to address this obligatory passage point and the need to form alliances in order to achieve their own

interests. Further, in defining the issue, the Managing Director needed to demonstrate her indispensability because the solution could only be reached through her (Callon 1986b, p. 204). Along with the issue, she was an obligatory passage point to the solution. The problematization of achieving systems integration creates a system of alliances between the entities. In order to forge an alliance between the entities involved, each of them needed to wish to respond to her SIN problematization as each actor had an interest in achieving a certain goal.

The IT team was crucial to the integration process and its role was difficult to achieve, increasing risks, because its members were given only a few months to accomplish the task. Furthermore, the acquired company used a very different IT system with different product codes, stock descriptions, and inventory identification numbers. The IT team was instrumental in converting all that information to Pronto style and then to integrating it with current files. Team members were concerned about the lack of compatibility of the two systems and time constraints, and saw these as significant obstacles. The IT manager provided an example showing the effect of an obstacle such as the resistance to systems integration:

> "The problem with that [systems integration] and all the company [acquiring company] is when we buy someone [e.g name] who runs in Exonet or MYOB they don't have the proper details such as name and address, post codes, correct product codes, proper description and everything else . . . it is a lot of work to get the data managed and get it into Pronto. The systems are not compatible".

At that stage, for the acquiring company management and employees to respond to the Managing Director's call, the personal relations between actors and agents needed to change and new alliances to emerge. The executives knew the importance of systems integration and its risks, but they were concerned about possible unpredicted Pronto implementation costs that might emerge later. Nevertheless, they were keen on future increased revenues and financial benefits as an outcome, so, they started sharing the same interests with the Managing Director and were willing to participate. On the other hand, the acquiring company employees recognised the unfamiliarity of the acquired company employees with Pronto. They were concerned this might mean additional work load for them as they had to train such employees. At that stage, the initial integration cost budget[4] (ICB) was produced. The Chief Executive Officer described it as:

> "In our investment recommendations we do make estimates on integration costs. They are very important in the process. Most of the businesses that we acquire are smaller business so the integration is not a significant, what needs an effort we use our internal resources, so we are not bringing external resources to do the integrations . . . it is done upfront before we acquire the business, we do look at the budget within acquisition cost, we do an analysis of the cost of integration and rationalization. We do that through due diligent process by getting exactly what we will bring across or we will not bring across".

Guided by its economic incentive figures, the Managing Director sought to tie together the divergent interests of these actors (Chua 2007) and get them to respond to her obligatory passage point. According to Callon (1986b):

> "Interessement is the group of actions by which an entity attempts to impose and stabilise the identity of the other actors it defines through its problematization. Different devices are used to implement these actions" (pp. 207–8).

The Managing Director explained:

> "I talked about moving them to a much larger brand-new building, better working conditions, providing modern separate offices for most of the employees, brand new office furniture, meeting rooms, kitchen facilities, computers, and privileges such as cars, lap tops and mobile phones".

Next, they created new roles, hired additional staff members and distributed the work load more effectively. Lunch and refreshments were made readily available during briefings and site meetings. Warehouse staff members were also well accommodated, with offices designated to administrative staff members inside the warehouse. All these appealed to the employees and gave them a sense of self and identity. They started to see the new vision and aspirations of 'what the new integrated acquiring company would look like'. They saw the company as a good place to work as it provided them with stable employment, easier operations, and was looking after their interests.

The Managing Director knew that the participation of the acquired company management was crucial for a successful integration, as their personal interests would influence the construction of SIN. Evidence shows, however, that the acquiring company management failed to recognise at the time of due diligence the significance of a key issue affecting the acquired company's management team. As the time passed, however, the acquiring company's management came to realise that the acquired company management did not wish to participate in integration efforts due to religious issues. The Managing Director explained:

> "[name] is quite a complex business, it was run by the Brethren, so obviously the owners would not work for any other company, it is not in their religious beliefs to work for non-brethren organization. So, they stayed with us until I think end of August, and should have been good enough for us to hand over and take control but obviously . . . the big thing about the Brethren is we believe that what they do is that they have gone and given all our intellectual property that we may have acquired to another brethren company. So, therefore, the [name] customers starting to be built up again, but certainly with the [name] customers there is a quite large portion of their major customers were brethren companies. Now during due diligence, we did ask the managing director [ . . . ] you know what impact this will have that we are non-brethren and they are brethren companies, what impact will that have in our sales? He said absolutely no impact what so ever. But of course, once they left, all those brethren companies stopped buying from us. However, what we have said to them is, we believe that you were dishonest in your disclosure to us during the due diligence and we have lost this business and therefore . . . They have not come back to us. I don't think I will ever buy a brethren business again . . . they were untrustworthy".

The Company Secretary described the situation:

> "This is quite unique to the industry [packaging], as it is based on a religious group referred to as 'Brethren', . . . they have certain beliefs that I am not totally familiar with it but from what I understand they would avoid things like computers, the modern day technologies, etc . . . and they also do a lot of business amongst themselves . . . which was the case in the [name]business. We were aware of it upfront, and we did do quite a lot of probing to ensure that this business would continue once under the new ownership".

The acquired company management team's strong belief that they could not work for a non-Brethren organisation led them to establish a new similar business, successfully encouraging a number of employees to join them. So, to the disappointment of the acquiring management, they demonstrated a total lack of interest, were not helpful in any of the integration efforts, and left the acquiring company later in the first year of acquisition.

The Chief Executive Officer stated:

> "I think the critical factors [in the success of integration] would be the fit of the vendors, vendors play a huge role in the way they come across, and clearly we did not have in [name]case, the cooperation and the way to make it succeed, the way we had with the others, so that is certainly a key element . . . they [name management] kept separate, they did not agree to integrate, they did not pass the knowledge on. So there was a lot of lessons we learned from that.

Hard lessons we learned from that acquisition, probably the worst, the least successful acquisition out of all the acquisitions that we have done".

The Managing Director explained:

"There were difficulties with whole integration, with the whole movement, so you know as I say, we underestimated, the way we understood the functions of some of those people [management of name] . . . We thought that they would be easily absorbed to our current infrastructure. I think we totally underestimated the extent of the involvement of these people, of these family members in the business".

The influence of the acquired company's management on their employees was great and they shared very strong bonds. The Warehouse Manager described the situation:

"The people who were in [name] warehouse, were a good team. They were also very much driven by their management. They were very close and were told exactly what to do, and then after you do that, then they would be doing something else. If you did not tell them they would stop. But that was because the senior guys from the [name] were Brethren, from a religion background so it always came from that".

In addition to their strong bonds with the 'Brethren' management team, the employees of the acquired company were facing another problem. They had not used Pronto previously and were not familiar with its features. So, using this system was challenging and they had to make an effort to learn the system to be able to use it on a daily basis. These obstacles caused issues for the Managing Director and the acquiring company's management when they attempted to connect with such staff in support of the SIN network. The problems that the acquired company's employees were facing needed to be addressed. Their participation was crucial to networking efforts, either as actors who work in the post-integration network or as actors who would pose no resistance to its formation. Mindful of these problems, the Managing Director explained the mandatory nature of the use of Pronto to them, and for it to be the major requirement to passing the Managing Director's obligatory passage point. They were promised employment and Pronto training. The Managing Director explained:

"At that stage, there was no funds, it was just slotted into our employee's daily work . . . so they would come in provide in house training for them... we probably paid overtime and they had to do it outside hours, so, that would have been budgeted in terms of overtime, but there was no specific training allocation".

The National Human Resources Manager further explained:

"We offer in house training and that is what happens . . . say we hire someone and they sit in customer service, ok, they get the overall induction, but we get the most experienced customer service person to train that new person, and that is how it works, and sometimes you know as time goes on, then that new person becomes very experienced and then they can take on the role of training".

A list of the entities, the obstacles and problems they faced and their goals that were involved in the SIN network are summarised in Figure 1. At this stage, the SIN network and the Managing Director as its representative were far from stable. Callon (1986b) states:

"It is open to challenges (known as 'trials of strength') from other entities (known as 'counter-actors') with their own agendas (known as 'anti-programmes'). While these trials of strength can occur at any time during the translation process, it is at the earlier phases when the network is still forming where relationships have not yet been tested, that it is most vulnerable" (pp. 207–8).

Subsequently, despite the concerns and lack of having the support of all the actors, the IT team were asked to commence its tasks. A project plan was prepared, detailing

tasks and responsibilities and due dates, and the team was to track against that. The initial stage of the plan involved a rigorous acquired company's product analysis, followed by suitable formatting. However, the IT department found it difficult to examine the existing acquired company's data in their systems, because, by that time, some key employees had left, and the remaining employees did not possess enough knowledge to assist the process.

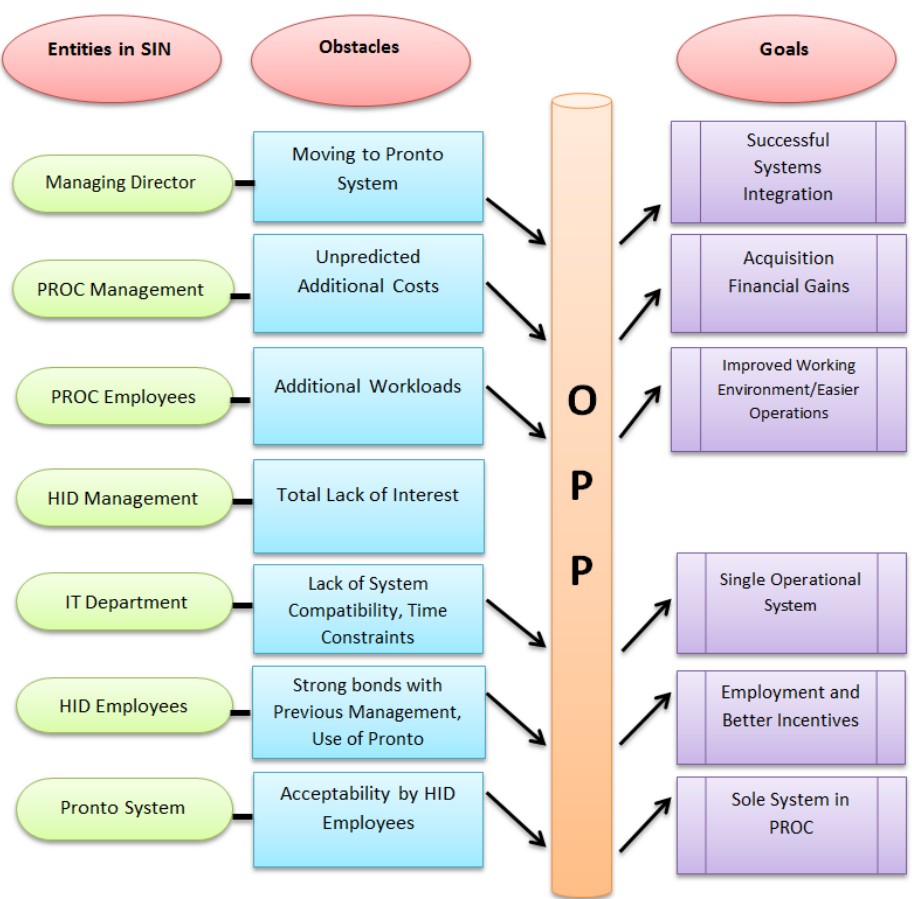

**Figure 1.** Systems Integration Network entities, their obstacles, OPP[5] and intended goals.

The next stage was that the conversion processes and procedures were automated and converted to Microsoft Excel and uploaded into the central system. So, the acquired company's products and customers, purchasing and sales, were integrated into Pronto. Unfortunately, most of the interviewees believed that systems integration was rushed and, indeed, it proved to be a costly and ineffective exercise. As the Managing Director explained:

> "I think it [systems integration] was pretty rushed conversions because we went through such a mess with our customer service perspective afterwards".

### 4.4. Stabilising the Systems Integration Network

At this stage, even though the powerful interessement device was used by the Managing Director, it was now up to the entities in the network to bind together and become enrolled in this process on the basis of the strategies being adopted by the focal actor." Callon (1986b) observes:

> "The mere creation of a network through the use of power does not ensure successful achievement of network goals' (p. 211).

At that point in time, some of the interessement attempts made by the Managing Director seemed unsuccessful, so the enrolment of all the actors was not achieved. Callon (1986b) describes the third moment of the translation framework:

"It is the process of binding the entities of a network together to work towards the achievement of the goal revealed in the moment of problematization. During enrolment, the problematizing actor is striving to transform what has been proposed into something that is more solid and certain" (p. 211).

In the next few months, a sales budget variation report[6] (SBV) was produced, displaying a large discrepancy between the predicted and actual sales budgets, updated sales figures, net income, price earnings ratio and EPS, in comparison with their competitors in the previous twelve months. This information was heavily used by the Chief Financial Officer to point out that the company was underperforming relative to its competitors. This report triggered a new level of contention in the organisation; people started arguments, blaming each other, and new political and social alliances emerged, leading to a loss of motivation and commitment to the SIN.

Subsequently, a team of external experts investigated the system problems, and found, since the conversion process in the initial stage was rushed, the implementation took place too swiftly, products were not integrated properly, and these presented as new problems in the warehouse. The Warehouse Manager explained:

"When they [name] came on Pronto there was a few duplications, few issues with codes, cross references and units of measures was not set up properly, it was such a whole new training thing for myself. Because the acquiring company side of things, for example, we sell a box of gloves, or sticky tape we sell it by the box, [name]actually would open the box and sell you one roll. So, when they integrated the [name]side of the products units of the measures were not set up properly, so sometimes customers would get extra and sometimes customers were getting less. And then also training the guys who are picking you have to correct the unit of measure, because they are so used to picking up the whole box you know. When they see something is coming through, for example a role of tape they question it. Because some of the key people had left we did not have the knowledge, there was a delay and timeframe to get it all up and to run it properly".

Furthermore, the team discovered some of the problems were due to differences in the two organisations' product handling procedures. Previously, the warehouse procedure was based on the location of the product. So, when an item was booked, the location was recorded in the invoice and picking staff could easily locate the product and pick it. The acquired company's systems had no feature to show the location of the product, and picking staff had to physically walk through many racks to find the product. This was time consuming and frustrating for employees. The Warehouse Logistics Manager described the problem:

"So, we had the warehouse segregated, so that was [name] stock and this was ours. The guys were getting sick of this system, and just did not know where to go to pick it. If one of the[name]guys trying to get acquiring company's people to go and pick it they would have got no clue, because they did not know where the things were. There was no marking, no location, the same as acquiring company ... ".

Another executive stated:

"Inventory related tasks such as product management, purchasing, distribution and transportation were not taking place efficiently and as a result there were significant delays in deliveries, wrong stock and quantities were being delivered and many customer issues emerged. So, in a short period of time, many of the [name] customers were lost and sales dropped significantly".

All these problems further weakened the SIN and resulted in what the Warehouse Manager explained:



"Frustration, resentment and even hostility started to build up in warehouse employees, and it was expressed in subversive behaviour revealing a loss of faith and trust in the competency and credibility of manager, some of the employees even were openly questioning Managing Director's decisions".

To make matters worse, due to the creation of new roles and additional remuneration packages offered to staff, financial resources were stretched and, as mentioned previously, cost budgets included minimal provisions for professional Pronto training. Consequently, in-house training that was offered produced additional unwelcomed workload on top of the employees' daily tasks and with no incentives. This led to inadequate training of the acquired company's employees, leading to major issues (such as delays in order processing, delivery, and customer service) in daily operations of the sales department and the warehouse. These issues were not addressed promptly and adequately, and became challenging and stressful for employees. Some of them felt excluded and not listened to, and a few, such as the Logistics Manager, felt that their job was expanding quite rapidly and they could not keep up. At that stage the Managing Director's attempt to enrol the actors to SIN faced great challenges, as it seemed that a number of other "actors attempted to translate the interests of the other actors" (Law 1992, p. 206). In other words, mutual translation occurred, as Latour (2005) describes:

"Translation is considered from the point of view of multiple actors simultaneously attempting to translate each other's interests into the network they build" (p. 111).

Callon (1986a) argued:

"Once a network is formed, however, that is not the end of the story as networks are always unreliable and can become unstable. The entry of new actors, desertion of existing actors or changes in alliances can cause the 'black-boxes' of networked actors to be opened and their contents reconsidered. A network relies on the maintenance of its simplifications for its continued existence. These simplifications are under constant challenge and if they break down, the network will collapse, perhaps to re-form in a different configuration as a different network" (p. 68).

The following paragraph discusses three instances of mutual translations where a number of other obligatory passage points were established to enrol the actors into relations. Discussing these instances is helpful to exploring the challenges faced by the initial obligatory passage point established by the Managing Director, and the existence of power relations in the network.

The first instance was when the acquired company's warehouse manager, who felt uncomfortable using Pronto, started a parallel translation by making negative comments and conversing with other unhappy acquired company employees, discouraging them from learning the system. He compared Pronto and Exonet and pointed out how the latter was more user-friendly and how training was inadequate. Then he resigned, left the acquiring company, and joined the newly established business formed by former acquired company vendors (re-joined the acquiring company a year later). He said:

"At that time with everything, moving and stuff, everything was all over the place, to be honest I just really got over it, and I did not want to work in such environment, so I went . . . ".

The remaining staff members lost their initial interest and excitement, as they felt frustrated by failing to use the system. A number of key employees followed him. The National Compliance and Logistics Manager explained the situation:

"I suppose when change is quite chaotic and rapid you might not be able to spend as much time with them [employees], making sure everyone is settling as they should be, so, you know you train them, you think they are ok, the trial, you let them run and you have got so many things to do and may be you are not

seeing what is wrong with people and that is when things start going off track . . . few people left at that time because they felt it did not fitted them".

Second, one of the acquired company managers held an informal meeting with all acquired company staff and expressed his negative views that the acquiring company's management was putting them under unnecessary pressure to disclose their customer sales contracts immediately. He encouraged the staff to refuse to co-operate, as he believed this was not part of the deal (when he knew it was). This attempt provoked a resurgence of old ties and created feelings of a lack of participation among acquired company staff. A warehouse employee said:

"It felt this was an intended exertion of power [name]'s people and had nothing to do with the achievement of better results and led to deepen tensions between us and acquiring company people".

Third, one of the acquiring company's senior managers (X) in a heated argument raised his concerns that systems integration was ineffective and causing problems in the functionality of the warehouse. He believed the capabilities of Pronto were not designed for integration of that scale and had to be extended, so he suggested that the acquired company run separately for the next twelve months until Pronto was ready. He demonstrated his change of interest by initiating another parallel translation attempt to steer the outcome of the resulting network to his own benefit, actively endeavouring to persuade everyone to his views and attempting to gain support. The struggle for efficient systems integration presented a challenge to the obligatory passage point established by the Managing Director as some of the executives, including the IT manager, agreed with his views. The Managing Director attempted to negotiate with X, but he would not agree to go through the obligatory passage point, and her efforts were unsuccessful. She then made a number of staff changes, including the replacement of X.

The following section describes how, finally, the initial obligatory passage point succeeded, strong relationships in the SIN were performed, and how power asymmetry was generated and deployed within the network.

Subsequent to all these new developments and mutual translations, the management team held a number of strategic meetings where there were lengthy discussions and exchanges of opinions. Finally, they made the decision to make fundamental changes to operational processes and procedures to improve performance. The first step was holding a meeting with the IT department Head, where he explained the problems and expressed his views that the current design of Pronto was incompatible with the acquired company's products. He suggested major updates and outlined necessary associating financial considerations. Surprisingly, he made a bold statement about his lack of willingness to be part of the systems integration under the current circumstances.

At the next meeting, a budget variation report[7] (BVR) was presented to the senior team and its unsatisfactory figures were distributed in hard copies. The team was greatly disappointed and they all felt decisive systematic action must take place, and members agreed to address all issues, including those raised by the IT Head. There were lengthy discussions, as any course of action required sufficient funds, which were not available after all the other considerations made previously. In the following days, the Managing Director met with one of the major wealthy shareholders (who was passionate and had a vision for the future of the company) and explained the goals of the integration and its future financial benefits, and requested his further investment. The Managing Director was pleased when he agreed to this request. Subsequently, the management team held further meetings, and reached an agreement on a new constructive project action plan detailing asks, responsibilities, and due dates for a proper successful systems integration.

However, there was a need for an updated cost budget[8] (UCB) to guide their actions. So, the Chief Financial Officer prepared an updated cost budget report that offered a generous budget, accommodating external resources and the IT Head requests. This budget contained a detailed outline of all the items and allocated funds, including Pronto updates

and training, hiring additional IT assistants, temporary external consultants, and supplementary incentives for the IT team's extra workloads and task completion. As he explained: *"to me the actual part at that stage was doing the budget and project plan, the physical part was pretty easy, that is not hard like we normally hit them on the dot"*. The IT Head was very pleased when he saw a copy of the updated budget that contained all his requested items. He sent a memo to the Managing Director and expressed his willingness to take on the task. The work of the focal actor as the initial translator was made interesting to this actor (the IT Head). Here the Managing Director acted upon the IT Head's interest, in such a way that he valued and utilised the network and became enrolled (Callon and Latour 1981). According to Callon and Latour (1981):

> "Numerous actors would have been targeted and pursued by the translator throughout processes of enrolment, but only some of these actors may qualify to be included in the establishment of the network. The third step in the enrolment process is, therefore, to establish convergences between the translatees and the translator and to use these as the basis for discriminating between actors. Establishing convergences entails determining homologies and equivalence between the interests, resources and strategies of the actors being translated and that of the translator. The translator is able to establish this by gathering information about the translatees, during the attribution and attraction processes, and transforming this information into equivalent values to enable him/her to establish various degrees of convergences needed to include or excludes translatees from the network. The process of establishing convergences in network formation creates conditions of possibility for management accounting" (p. 211).

The team led by the IT Head was formally assigned to migrate the acquired company's Exonet systems into Pronto. In the following days they hired a few additional assistants and commenced their work. They held a series of weekly meetings to address Pronto problems, eliminate duplications, and fix incorrect product units of measures. They worked tirelessly in the coming months.

Soon the Managing Director formed close alliances with other actors in the organisation and the integration team mobilised support for the problem (integration) constructed by the managing director and top management. Yet still, the old ties of a few acquired firm employees with Brethren management and lack of familiarity with Pronto was an obstacle. It presented a great challenge in aligning the entire network towards the interests of integration networks, related to an obligatory passage point through which all actors needed to pass. The first course of action taken by the Managing Director was to address this later obstacle. She employed external consultants on a contract basis to regularly provide extensive onsite training and immediate solutions to problems. They spent a lot of time and gave attention to the processes of the warehouse, providing hard and soft copies of 'User Guide Manuals' in several warehouse locations. These guides were written in very simple English that staff members could quickly access and easily read. Training followed with user guides and was a significant step in improving the functionality of the warehouse. In the next few weeks, many problems were identified and resolved. The consultants made an active effort to guide employees in using Pronto, and evidence showed that improvements followed. According to the Warehouse Manager: "With having trainers and system 'user manuals' on site, our productivity became tenfold, sales were smooth"!

In addressing the first obstacle of staffing, once again, guided by the figures of the integration cost report, the Managing Director took on the second round of negotiations with the acquired firm employees who had left the acquiring company. Having sufficient funds allocated for new employees, above market salaries, enabled her to better discuss positions. She made efforts to accommodate all the acquired firm employees. The Human Resources Manager said:

> "People became part of it (structure of the acquiring company), they got to keep, obviously where the skill set was, so got to utilise that and get to keep that. There

were trials, certain people going in different areas and if it worked it worked, if it didn't, we found another fit that would suit their skill set . . . what they do have access to I suppose, is a good remuneration package, I can guarantee you that everyone is paid above award, we work very closely with the individuals to make sure they are happy in their roles and that is both working for them and the company . . . ".

They showed interest and willingness to return and participate, and all signed employment contracts and commenced their employment. The Chief Executive Officer stated "I think in many cases they actually prefer to work for us now". It was very helpful when, subsequently, the former acquired company's warehouse manager returned and took up the position of Warehouse Receiving Supervisor.

### 4.5. Systems Uniformity

The last and Fourth Moment of translation is mobilisation. This moment is a type of monitoring where the spokesperson assures the silence of those others in the network being represented by the spokesperson (Dumay and Rooney 2016). Mobilisation is about ensuring that all the elements in the assembled network fulfil their assigned roles; Those who claim to speak for others truly do represent them by ensuring that none defects from the network or, if they do defect, none represents the network or spokesperson (Callon 1986b).

Two years after the acquisition, Pronto was being uniformly used across all the departments in the organisation. New employees with various roles were provided with adequate training, and new issues and problems were easily solved by referring to "User Guide Manuals'. It seemed that the Managing Director had achieved the acquiring company's strategic goal in integrating all the systems and successfully navigating through the original obligatory passage point. Updated budgets were successful in mobilising entities in the network through the interessement device of the integration network. Employees seemed satisfied with the support that they were getting and there were reports of improvement in revenue in the coming months, with surveys indicating higher performance and employee satisfaction rates. The following statement made by an employee demonstrates this:

"I love working here, I think this is a great company with a lot of potential, you know the range of product that we have is unbelievable and I think we need to embrace that and move forward rather than being caught in the past".

At the strategy meeting held later in the second year, the Chief Financial Officer, pointed out to the senior executives the article in the *Financial Review* featuring the company and reporting its good financial performance, and indicated that having a vision of A$1 billion revenue presented new possibilities and strong publicity for the company. He said there should probably be upcoming articles reporting much higher performance of the company in the near future, leading to additional investors. The researcher left the field around the time that reports of improvement in the sales and customer satisfaction emerged and when there were indications of exploring further acquisition opportunities, with lessons learnt from the acquisition reported in this paper. Outcomes of her investigation and analysis, grounded in the research interviews, are summarised in Figure 2 below.

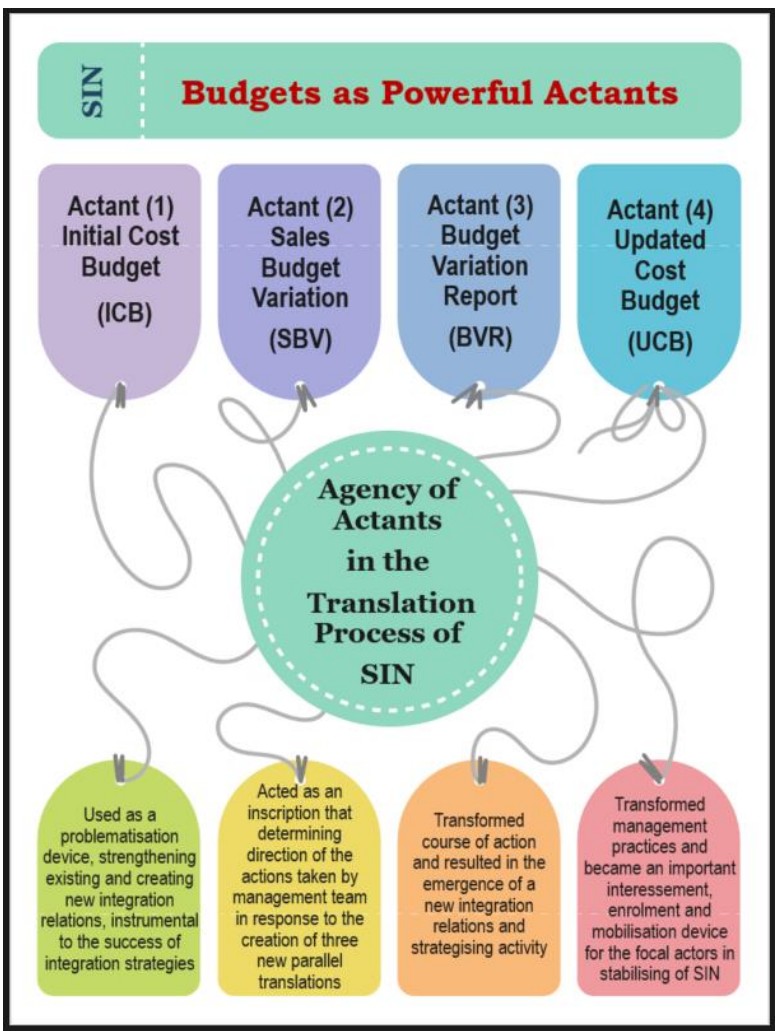

**Figure 2.** Summary Results.

## 5. Discussion

Analysis of this case demonstrates that the Managing Director used her power to make the use of Pronto mandatory and constitute SIN. The power of the Managing Director as the focal actor in developing systems inscription is based on her "in potentia" (Latour 1986), a power that is gained through organisational authority. This power allowed her to direct other actors in the network to obey her commands. For instance, the Managing Director used her power to obtain additional much needed funds from a wealthy shareholder to support improved systems integration activities, or when she accommodated the IT Head's request to update Pronto capabilities, or when she replaced X.

But, this was not the main element that played a significant role in the creation of SIN. The Managing Director's 'in potentia' powers were complemented by the "in actu" (Latour 1986, p. 265) power displayed by budgets as the crucial non-human actant. This was the power referred to by Latour as the power to "influence the actions of others". Budgets had no power 'in potentia', but in the interaction with human actors (such as the Managing Director and IT manager) the gained 'in actu' power was successful in translating the important actors into successful SIN.

The updated sales budget presented by the Chief Financial Officer laid a performative role and generated important information on the sales figures that enabled the management team to see the exact amount of discrepancy between what was expected in sales and what was achieved, clarifying the risk dimensions. It provided a clear quantification of insufficient performance of the company in comparison to rivals. Disappointing sales

figures demonstrated by the accounting report made executives see the reality and grasp a clear understanding that ineffective systems integration led to the situation where the expected synergies, such as enhanced revenue and cost savings from the acquisition, had not eventuated, and, consequently, the need to work together to find a solution. The Sales budget as an act of engagement made interactions (Law 1992) among actors possible. This was evident when a new set of actions emerged, such as a fresh 'strategic action plan' or the decision to appoint a group of experts to investigate the systems issues and identify problems (Hopwood 1973; MacKenzie 2006). It also led to additional unplanned and unexpected action (Davila 2005; Razi and More 2017) where the Managing Director decided to approach the wealthy shareholder to obtain the necessary funds to enable the executive team to implement their new strategy.

Furthermore, the information on the 'lost sales' figures (as a consequence of lack of familiarity of the acquired company's staff with Pronto) was important and led the management team to take a new action by making Pronto training a priority at that later stage in the change process. They became involved in new action where they allocated funds and hired external consultants and additional IT assistants to assist with systematic training and solving system integration issues. The outcome of this was the emergence of the new inscription 'User Guide Manual' (Callon 1986b) and new practice of the uniform use of Pronto across the organisation as summarised in Figure 2.

Another notable issue is that the influence of the acquired company's management "changed the interest and nature of relationships in the network and destabilised the formation of alliances in support" (Latour 1999, p. 123) of the integration vision. However, as integration efforts progressed, *'in actu'* powers of budgets inspired human actors to disintegrate the network of alliance between Brethren and employees and align the interests of the acquired company employees with the focal actor once again. Despite the initial mutual translation, risks and challenges faced, at the end, the diverse interests among actors were weakened, and common interests and mutual goals were established (MacKenzie 2006).

As outlined in Figure 3, budgets played a high degree of influence over the actors, being an effective interessement and enrolment tool that was deployed by management to persuade and enrol actors into enactment of integration strategies and formation of SIN. They played a critical role by transforming the information into convergent values that determined the composition of the SIN network. This acted as the "network builders" (Callon 1986b, p. 123), and became central to the expertise of management and employees who contained knowledge of integration practices.

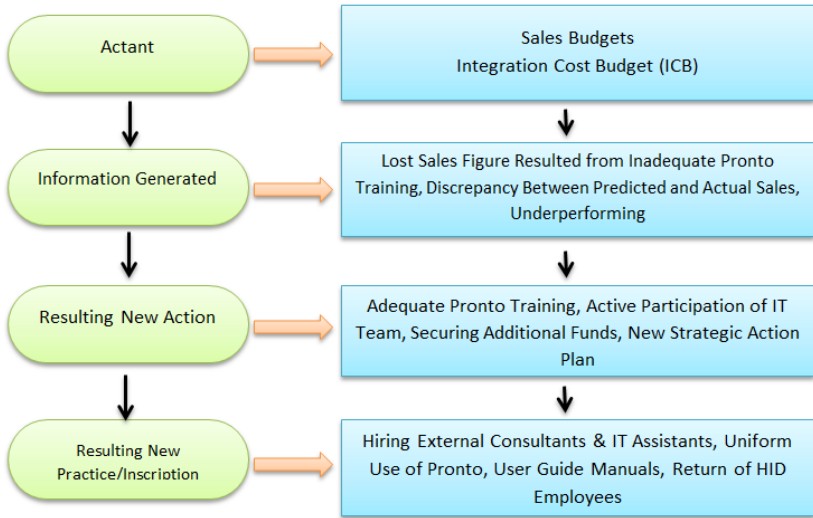

**Figure 3.** Performative role of budgets in the acquiring company systems integration.

## 6. Conclusions

This case study examined the performative role of budgets as the control system in implementing the change process of post-acquisition systems integration and reducing its associated risks in an Australian acquisition case. By drawing largely on the framework of Callon (1986b), translation and Actor Network Theory, the study has identified a number of useful themes that make both theoretical and empirical contributions to accounting, change, and risk literatures.

The case study demonstrated the role of human actors in the post-acquisition period by the facts showing that the essence and form of the risky integration strategy was determined by the ties which connected it to the integration activities of diverse actors, often located in many places. The integration strategy had a variable ontology. It obtained its forms as a consequence of the relations in which it was located, best understood as bundles of integration activities, which emerged in and through actor-networks, often in an unanticipated manner. Strategizing SIN is more or less a network effect and a process of co-production.

The case study emphasised the role of non-human factor such as the budget controlling human activity and action which cannot mistakenly be confined to human actors just because they are more visible 'on stage', given that non-human actors also make actors act (Callon 1986b). Budgets as a non-human actor, acted as a structuring and inscription tool that transformed the acquiring company's integration activities.

The case study highlighted that the identity and intentions of key strategic actors were highly dependent on budgets, playing a performative role in enacting and formulating integration strategy and its constitution. Systems integration networks were formed with limited exercise of power 'in potentia'. It was the budgets' power of 'in actu' (Latour 1986) that guided the actions of the executives and employees towards successfully achieving the acquiring company's strategic change goal and overcoming some of the deficiencies noted in the acquisition due diligence process.

The study explored the 'integration strategy' from a fresh point of view, noting its inherent risk, by proposing that an integration strategy can be thought of as a visionary object that does not exist in a concrete form and is not something that an organization can possess. Instead, it is best understood in processes of translation, that is, when the strategy themes are 'acted out', performed and translated through practical integration activities. An integration strategy comes to be defined relationally (Callon 1986b).

This study case, by examining accounting processes in action in creating SIN, contributed to a different narrative approach on the evolution of accounting and its effects on the risks of integration processes. Currently, there is inadequate evidence on the enactment of accounting in the practices of the often-challenging post-acquisition integration. The emphasis on studying accounting in action allows for investigation of the processes through the role of accounting in the change processes of acquisitions and SIN.

It should be acknowledged that the main limitations of this study in providing the account of systems integration networks is that it is a singular account, covering two Australian companies, in a particular industry. Further studies are needed to investigate findings of this research in other settings of companies, industries, and countries.

**Author Contributions:** All three authors contributed equally. Writing—original draft, N.R., Writing—review & editing, N.R., E.M. and G.S. All authors have read and agreed to the published version of the manuscript.

**Funding:** This research received no external funding.

**Institutional Review Board Statement:** Not applicable.

**Informed Consent Statement:** Not applicable.

**Data Availability Statement:** Data results can be find in Nazila Razi's Phd thesis. 2016 La Trobe University.

**Conflicts of Interest:** The authors declare no conflict of interest.

**Appendix A. Interview Guide**

*Appendix A.1. Personal Details*

How would you describe your current role in this organisation?

How long have you worked in the company? How long have you been in your current position?

What background and experience do you bring to this position?

Tell me about the types of work you perform in this company.

Did you work in the organisation at the time of M&A? What was your role before the acquisition? What is your role now? How has your work changed? How involved are you in acquisition decisions?

*Appendix A.2. Questions Related to Research*

Describe all the steps involved in the planning and implementation of post- acquisition integration (please describe it like a story from the beginning to date), covering the following:

What sort of management control systems were used throughout the integration? How they were used?

Which factors do you think have been most important in the success/failure of the post- acquisition integration?

Describe your role in the post- acquisition integration process.

What sort of constraining policies/actions have been introduced? Describe problems that arose as a result.

Are you satisfied with the integration performance? Why and why not?

What do you believe was done particularly well?

Do you think integration performance met its objectives? If not, what do you believe could have been done differently?

What do you think helped the integration success the most?

What were the most challenging issues in the integration processes?

*Appendix A.3. Describe Budgets within the Company*

Did you follow a specific budget for integration activities? Describe it. How were budgets used?

**Notes**

[1] This study investigated all aspects of the integration (systems, HR, customer, sales), but only systems integration analysis is included in this paper.

[2] Real names of both acquiring and acquired company are disguised for confidentiality reasons.

[3] D efine an inscription as follows: "the inscription is found at the end of a process of sorting out traces and connecting them in an expression.

[4] This is referred to as ICB in the Figure 1.

[5] Obligatory Passage Point (OPP) shown in Figure 1, can be defined as a "single locus that could shape and mobilize the local network" and "have control over all transactions between the local and the global networks."

[6] This is referred as SBV in the Figure 2.

[7] This is referred to as BVR in the Figure 2.

[8] This is referred to as UCB in the Figure 2.

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
