# Peer review of "Risk Implications for the Role of Budgets in Implementing Post-Acquisition Systems Integration Strategies"

_jrfm, doi:10.3390/jrfm14070323_

Round 1
Reviewer 1 Report
This is a very interesting piece of research that explores an under-explored research gap. A full case study is analytically presented and it provides an essential contribution to the research.
Company integration is analysed from an AIS system integration a budgeting perspective, however, it is not so clear in the abstract a specific identification of the research questions is missing.
Although the paper is well presented, structured and supported by a relevant theoretical framework, the title of the paper fails to describe the research. I encourage the authors to include some missing keywords such as "budget", as the accounting perspective is a major driver of the research.
In the introduction, I would also suggest including similar case studies of companies that struggled to integrate their systems for comparative analyses.
In the final paragraph of the conclusions only one generic limitation is disclosed. The author(s) might consider being more specific and try to identify other possible limitations.
In Figure 1 it will be better to disclose the meaning of "OPP".
Author Response
COMMENT 1: It is not so clear in the abstract a specific identification of the research questions is missing.
Response: PROPOSED NEW ABTRACT:
This paper studies the role of budgets in implementing the systems integration strategies in an Australian post-acquisition case of two organisations and reducing its associated often-regarded high risks.
It attempts a fresh narrative approach to examine the evolution of accounting and its effects on the challenges of post-acquisition integration processes by using the performative approach, such as the sociotechnical networks of Actor Network Theory, in a broader analytical framework as a possible solution to reducing the risks inherent in systems integration.
The methodology of the case study is based on Callon’s (1986) model of Four-Moment translation where integration strategy and budgets are regarded as social practice and defined relationally as bundles of activities and take form in and through practice and interaction between diverse actors and actants. A qualitative approach is adopted in the examination of the systems integration networks in an Australian post-acquisition case. Data was collected and analysed using semi-structured interviews.
It is found, through the examination of the routine practices of systems integration strategy making and how people enact and draw on a certain financial report on a daily basis to perform systems integration network strategies, that material forms of accounting act as a powerful structuring and inscription tool in integration activities, thus shaping integration strategic options and post-acquisition economic conditions of the organisation. The result shows how the risk could be reduced in post-acquisition system integration.
The research contributes to the risk, change and accounting literatures by providing insights into the mundane and ordinary practices of different aspects of integration strategy making, and the way employees enact and draw on accounting numbers on a day-to-day basis to perform systems integration network strategies. The case study facilities this research being further developed and broadened in terms of other cases, industries and countries.
Keywords: Systems Integration Networks, Budgets, Actor Network Theory, human and non-human actors, Post-acquisition risks.
COMMENT 2: The title of the paper fails to describe the research.
Response: Proposed New Title: Risk Implications for the Role of Budgets in Implementing Post-Acquisition Systems Integration Strategies
COMMENT 3: I encourage the authors to include some missing keywords such as "budget", as the accounting perspective is a major driver of the research.
Response: The word ‘budgets’ is included.
COMMENT 4: In the introduction, I would also suggest including similar case studies of companies that struggled to integrate their systems for comparative analyses.
Response: Two new articles (YouLegal, 2019) and (Investopedia, 2021) on some failures in mergers and acquisitions cases (USA and Australia) as a comparative reference to the difficulties in systems integration are included.
COMMENT 5: In the final paragraph of the conclusions only one generic limitation is disclosed. The author(s) might consider being more specific and try to identify other possible limitations.
Response: Proposed New Conclusion
This case study examined the performative role of budgets as the control system in implementing the change process of post-acquisition systems integration and reducing its associated risks in an Australian acquisition case. By drawing largely on the framework of Callon (1986), translation and Actor Network Theory, the study has identified a number of useful themes that make both theoretical and empirical contributions to accounting, change and risk literatures.
The case study demonstrated the role of human actors in the post-acquisition period by the facts showing that the essence and form of the risky integration strategy was determined by the ties which connected it to the integration activities of diverse actors, often located in many places. The integration strategy had a variable ontology. It obtained its forms as a consequence of the relations in which it was located, best understood as bundles of integration activities, which emerged in and through actor-networks, often in an unanticipated manner. Strategizing SIN is more or less a network effect and a process of co-production.
The case study emphasised the role of non-human factors such as the budget as controlling human activity and action which cannot mistakenly be confined to human actors just because they are more visible ‘on stage’, given that non-human actors also make actors act (Callon, 1986). Budgets as a non-human actor, acted as a structuring and inscription tool that transformed the acquiring company’s integration activities.
The case study highlighted that the identity and intentions of key strategic actors were highly dependent on budgets, playing a performative role in enacting and formulating integration strategy and its constitution. Systems integration networks were formed with limited exercise of power ‘in potentia’. It was the budgets’ power of ‘in actu’ (Latour, 1986) that guided the actions of the executives and employees towards successfully achieving the acquiring company’s strategic change goal and overcoming some of the deficiencies noted in the acquisition due diligence process.
The study explored the ‘integration strategy’ from a fresh point of view, noting its inherent risk, by proposing that an integration strategy can be thought of as a visionary object that does not exist in a concrete form and is not something that an organization can possess. Instead, it is best understood in processes of translation, that is, when the strategy themes are ‘acted out’, performed and translated through practical integration activities. An integration strategy comes to be defined relationally (Callon, 1986).
This study case contributed, by examining accounting processes in action in creating SIN, to a different narrative approach on the evolution of accounting and its effects on the risks of integration processes. Currently, there is inadequate evidence on the enactment of accounting in the practices of the often-challenging post-acquisition integration. The emphasis on studying accounting in action allows for investigation of the processes through the role of accounting in the change processes of acquisitions and SIN.
It should be acknowledged that the main limitations of this study in providing the account of systems integration networks is that it is a singular account, covering two Australian companies, in a particular industry. Further studies are needed to investigate findings of this research in other settings of companies, industries and countries.
COMMENT 6: In Figure 1 it will be better to disclose the meaning of "OPP.
Response: A foot note is included at the end of the page 14 to address this.

Reviewer 2 Report
The research study the role of budgets in the systems integration strategies of two organisations. For this purpose, the authors use a qualitative approach,i.e., semi-structured interviews.
The paper is very interesting, I enjoyed the literature review and discussions. However, the current state of the research has some important gaps.
1) The title is generic, while the authors analyze only the Australian market. I suggest changing it
2) Abstract. I understand the " free format submission" option, but the abstract is structured like the Emerald journals. I suggest changing it. It is not in line with the journal
3) Introduction. It is well structured, but I think the authors should focus mainly on why it is important to study this relationship in the Australian context since that is what they are analyzing,
4) Section 3. The methodology is not well explained in the sense that there are important gaps: why was this sample chosen? What methods of sample selection? Random sample? Why 20 employees?
5) All of Section 3 seems like a mere summary of the meeting and various employee comments. I recommend that the authors edit this section by making it more streamlined. I also recommend showing the interview data, i.e. the questions.
6) Throughout the paper, there are too many "paragraphs" of Callon's research. I understand the importance, so doing so risks making a summary of his work.
7) The results are presented as "generic", but they refer only to the Australian context for 20 workers. I suggest the authors contextualize their results (discussion and conclusions)
Author Response
Response to Reviewer 2 Comments
COMMENT 1: The title is generic, while the authors analyze only the Australian market. I suggest changing it.
Response: Proposed New Title: Risk Implications for the Role of Budgets in Implementing Post-Acquisition Systems Integration Strategies
COMMENT 2: Abstract. I understand the " free format submission" option, but the abstract is structured like the Emerald journals. I suggest changing it. It is not in line with the journal
Response: PROPOSED NEW ABTRACT:
This paper studies the role of budgets in implementing the systems integration strategies in an Australian post-acquisition case of two organisations and reducing its associated often-regarded high risks.
It attempts a fresh narrative approach to examine the evolution of accounting and its effects on the challenges of post-acquisition integration processes by using the performative approach such as the sociotechnical networks of Actor Network Theory in a broader analytical framework as a possible solution to reducing the risks inherent in systems integration.
The methodology of the case study is based on Callon’s (1986) model of Four-Moment translation where integration strategy and budgets are regarded as social practice and defined relationally as bundles of activities and take form in and through practice and interaction between diverse actors and actants. A qualitative approach is adopted in the examination of the systems integration networks in an Australian post-acquisition case. Data was collected and analysed using semi-structured interviews.
It is found, through the examination of the routine practices of systems integration strategy making and the way of how people enact and draw on a certain financial report on a daily basis to perform systems integration network strategies, that material forms of accounting act as a powerful structuring and inscription tool in integration activities, thus shaping the integration strategic options and post-acquisition economic conditions of the organisation. The result shows that the risk could be reduced in the post-acquisition system integration.
The research contributes to the risk, change and accounting literatures by providing insights into the mundane and ordinary practices of different aspects of integration strategy making, and the way of how employees enact and draw on accounting numbers on a day to day basis to perform systems integration network strategies. The case study facilities this research to be further developed and broadened in terms of cases, industries and countries.
Keywords: Systems Integration Networks, Budgets, Actor Network Theory, human and non-human actors, Post-acquisition risks.
COMMENT 3: Introduction. It is well structured, but I think the authors should focus mainly on why it is important to study this relationship in the Australian context since that is what they are analyzing,
Response: Five new lines are included after the third line as follows: In 2018, Australia experienced 1900 M&As, with a value of approximately $185 Billion with many of them failing due to the fact that risks were not fully understood (YouLegal, 2019). For instance, Wesfarmers acquired Coles in 2007 for $21b. But recently Wesfarmers demerged from the Coles Group and both are back where they started. So, it is important to study this in the Australian context.
COMMENT 4: Section 3. The methodology is not well explained in the sense that there are important gaps: why was this sample chosen? What methods of sample selection? Random sample? Why 20 employees?
Response: Five new lines are added at the start of this section as follows:
Australian Stock Exchange (ASX) listed companies that were engaged in M&A activities were closely examined and potential candidates for the purpose of this study were identified. Among those a packaging company (PROC[1]) as a manufacturer and distributor of packaging products was chosen for this study. This particular sample was chosen because this company made an interesting case study for the purpose of this research and represented a significant part of Australian industry that has grown even more during the COVID pandemic recently.….
ALSO: The following clarification has been included at the end of the page 8: The main body of the data was collected through semi-structured interviews agreed by the organisations involved as lasting approximately for one hour each. Crucially, all individuals who were directly involved with the acquisition and integration activity (20 in total) were interviewed without exception. This enabled the researcher to gain richer data than perhaps other methods might have afforded.
COMMENT 5: All of Section 3 seems like a mere summary of the meeting and various employee comments. I recommend that the authors edit this section by making it more streamlined. I also recommend showing the interview data, i.e. the questions.
Response: The approach taken is standard in this type of research and analysis. Interview data is added at the end of the paper.
COMMENT 6: Throughout the paper, there are too many "paragraphs" of Callon's research. I understand the importance, so doing so risks making a summary of his work.
Response: The quote in page nine is removed, but removing more quotes affects the articulation of the arguments in the paper.
COMMENT 7: The results are presented as "generic", but they refer only to the Australian context for 20 workers. I suggest the authors contextualize their results (discussion and conclusions)
Response: Discussion and conclusions are re-written to address this.
[1] Real names of both acquiring and acquired company are disguised for confidentiality reasons.

Reviewer 3 Report
Dear Authors
Your article is very interesting, well-written and very clear, I am grateful for the opportunity to read it. I think that idea and the subject of the research are very interesting, and the results of the research give a lot of new information and possibilities of further analysis. Overall, I am very impressed because your article is well-prepared.
Reading the text, I found only 1 element that I think would improve your article.
The Introduction and Conclusions lack the formal elements of your article. In my opinion conclusions are insufficient - The conclusion section should be a summary of article’s aim, methods and findings. But it's not here. This chapter should be extended. For me, the summary is too limited, there is no reference to your assumptions/hypotheses/research questions. At this point, you should show references to your research and all formal aspects of your article. At the begging (introduction) and at the end (Conclusion) you should include a description of the research questions and/or research hypotheses. You should develop and explain your goals. It is necessary to change the convention from the presentation of research to the presentation of results and conclusions. In general, I believe that hypotheses and research questions should be presented. The goals/assumption should be presented and explained. And at the end, the conclusions should refer to each of the goals and research questions.
Summarizing.
I find your article very good. I really like your article and appreciate your work. It is interesting topic, and the conclusions could open the way for further research. You have to make some changes in Introduction and Conclusions. But my general opinion and my assessment of your research and whole article is more than positive.
Good luck!
Author Response
Response to Reviewer 3 Comments
The Introduction and Conclusions lack the formal elements of your article.
COMMENT: In my opinion conclusions are insufficient – The conclusion section should be a summary of article’s aim, methods and findings. But it's not here. This chapter should be extended. For me, the summary is too limited, there is no reference to your assumptions/hypotheses/research questions. At this point, you should show references to your research and all formal aspects of your article. At the begging (introduction) and at the end (Conclusion) you should include a description of the research questions and/or research hypotheses. You should develop and explain your goals. It is necessary to change the convention from the presentation of research to the presentation of results and conclusions.
Response: Conclusion section is re-written as follows:
This case study examined the performative role of budgets as the control system in implementing the change process of post-acquisition systems integration and reducing its associate risks in an Australian acquisition case. By drawing largely on the framework of Callon (1986), translation and Actor Network Theory, the study has identified a number of useful themes that make both theoretical and empirical contributions to accounting, change and risk literatures.
The case study demonstrated the role of human actors in the post-acquisition period by the facts showing that the essence and form of the risky integration strategy was determined by the ties which connected it to the integration activities of diverse actors, often located in many places. The integration strategy had a variable ontology. It obtained its forms as a consequence of the relations in which it was located, best understood as bundles of integration activities, which emerged in and through actor-networks, often in an unanticipated manner. Strategizing SIN is more or less a network effect and a process of co-production.
The case study emphasised the role of non-human factor such as the budget as the control over human activity and action which cannot mistakenly be confined to human actors just because they are more visible ‘on stage’, given that non-human actors also make actors act (Callon, 1986). Budgets as a non-human actor, acted as a structuring and inscription tool that transformed the acquiring company’s integration activities.
The case study highlighted that the identity and intentions of key strategic actors were highly dependent on budgets, playing a performative role in enacting and formulating integration strategy and its constitution. Systems integration networks were formed with limited exercise of power ‘in potentia’. It was the budgets’ power of ‘in actu’ (Latour, 1986) that guided the actions of the executives and employees towards successfully achieving the acquiring company’s strategic change goal and overcoming some of the deficiencies noted in the acquisition due diligence process.
The study explored the ‘integration strategy’ from a fresh point of view, by proposing that an integration strategy can be thought of as a visionary object that does not exist in a concrete form and is not something that an organization can possess. Instead, it is best understood in processes of translation, that is, when the strategy themes are ‘acted out’, performed and translated through practical integration activities. An integration strategy comes to be defined relationally (Callon, 1986).
This study case contributed, by examining accounting processes in action in creating SIN, to a fresh narrative approach on the evolution of accounting and its effects on the risks of integration processes. Currently, there is inadequate evidence on the enactment of accounting in the practices of the often-challenging post-acquisition integration. The emphasis on studying accounting in action allows for investigation of the processes through the role of accounting in the change processes of acquisitions and SIN.
It should be acknowledged that the main limitations of this study in providing the account of systems integration networks is that it is a singular account. Further studies are needed to investigate findings of this research in other settings of companies, industries and countries.
COMMENT 2: In general, I believe that hypotheses and research questions should be presented. The goals/assumption should be presented and explained.
Response: The key Research question is clearly stated in the Introduction section (second last paragraph of page 2). Goals are explained in the second paragraph of page 7.

Round 2
Reviewer 2 Report
The authors have only partially modified their work. I appreciated the effort, but nevertheless, I still think the work needs a revision to be published.
The authors reply to my comment 3, including two not exactly academic citations (Investopedia; youlegal). I think the authors should change them. Furthermore, the authors should better emphasize their contribution and the choice of the Australian market. Don't be too generic.
Reply to comment 4. The authors have entered several comments. However, I find the choice of the Australian context unclear, i.e. the authors explain that they chose this market because "This particular sample was chosen because this company made an interesting case study for the purpose of this research and represented a significant part of Australian industry that has grown even more during the COVID pandemic recently ".
I think it's a too general response; every nation has had problems with the COVID-19 pandemic. I would like the authors to explain their reasons better.
Reply to comment 5. The approach is the standard one, but this looks like a summary of the meeting and the various comments. I would like the authors to be more "technical", i.e. for example, you could insert graphics to have a clearer and more complete view of the results. Unfortunately, in the attached file, there are no interview data.
Author Response
Response to the second round of Reviewer 2 comments
Feedback 1: The authors reply to my comment 3, including two not exactly academic citations (Investopedia; youlegal). I think the authors should change them. Furthermore, the authors should better emphasize their contribution and the choice of the Australian market. Don't be too generic.
Response: following paragraph has been added at the beginning of ‘Introduction’ section
The study on the issue of strategic integration in post-acquisition process is due to the rich literature reporting the failures of most acquisitions and mergers world-wide, including in Australia. For example, Yaw (2016) reported that “M & A failure rate is very high; averaging about 50%, regardless of the initial high hopes” and also that (p.1). “the integration stage as one of the critical stages within the whole M & A process which can contribute immensely to M & A failure”.
Feedback 2: Reply to comment 4. The authors have entered several comments. However, I find the choice of the Australian context unclear, i.e. the authors explain that they chose this market because "This particular sample was chosen because this company made an interesting case study for the purpose of this research and represented a significant part of Australian industry that has grown even more during the COVID pandemic recently ".
Response: Following has been added at the beginning of section 3
The study on the Australian cases of acquisitions and mergers is motivated by a number of factors. A crucial one is from a methodological perspective. Choosing the Australian context meant the researcher being able to do the face-to-face interviews and other in person research that was deemed necessary for the study envisaged. Additional factors included the nature of Australia’s small open economy with a long history of both attracting foreign investment and investing overseas (Australian Productivity Commission Research Report: Foreign Investment in Australia, June 2020), allowing study contributions in advancing Australia’s economic development. Moreover, the Australian focus was designed to facilitate comparative studies using the interesting work of Karagiannidis (2010) on the synchronisation of fluctuation in trends of acquisitions and mergers between Australian, USA and UK.
Reference list is updated accordingly.
Reply to comment 5. The approach is the standard one, but this looks like a summary of the meeting and the various comments. I would like the authors to be more "technical", i.e. for example, you could insert graphics to have a clearer and more complete view of the results. Unfortunately, in the attached file, there are no interview data.
Response: Appendix ‘A’ as interview guide is added at the end of the paper. A new graph (Table 3.1 Summary Results) is inserted at the end of section 3.

Round 3
Reviewer 2 Report
The authors modified the paper following my suggestions. I think the paper is now ready for publication.
all the best